# Impact on alcohol selection and online purchasing of changing the proportion of available non-alcoholic versus alcoholic drinks: A randomised controlled trial

**Natasha Clarke**[1,2]*, **Anna K. M. Blackwell**[3,4], **Jennifer Ferrar**[3], **Katie De-Loyde**[3], **Mark A. Pilling**[1], **Marcus R. Munafò**[3], **Theresa M. Marteau**[1]*, **Gareth J. Hollands**[1,5]*

**1** Behaviour and Health Research Unit, Department of Public Health and Primary Care, University of Cambridge, Cambridge, United Kingdom, **2** School of Sciences, Bath Spa University, Bath, United Kingdom, **3** School of Psychological Science, Tobacco and Alcohol Research Group, University of Bristol, Bristol, United Kingdom, **4** Department of Psychology, University of Bath, Bath, United Kingdom, **5** EPPI Centre, UCL Social Research Institute, University College London, London, United Kingdom

* n.clarke@bathspa.ac.uk (NC); tm388@medschl.cam.ac.uk (TMM); gareth.hollands@ucl.ac.uk (GJH)

## Abstract

### Background

Increasing the availability of non-alcoholic options is a promising population-level intervention to reduce alcohol consumption, currently unassessed in naturalistic settings. This study in an online retail context aimed to estimate the impact of increasing the proportion of non-alcoholic (relative to alcoholic) drinks, on selection and purchasing of alcohol.

### Methods and results

Adults ($n$ = 737) residing in England and Wales who regularly purchased alcohol online were recruited between March and July 2021. Participants were randomly assigned to one of 3 groups: "25% non-alcoholic/75% alcoholic"; "50% non-alcoholic/50% alcoholic"; and "75% non-alcoholic/25% alcoholic," then selected drinks in a simulated online supermarket, before purchasing them in an actual online supermarket. The primary outcome was the number of alcohol units selected (with intention to purchase); secondary outcomes included actual purchasing.

A total of 607 participants (60% female, mean age = 38 years [range: 18 to 76]) completed the study and were included in the primary analysis. In the first part of a hurdle model, a greater proportion of participants in the "75% non-alcoholic" group did not select any alcohol (13.1%) compared to the "25% non-alcoholic" group (3.4%; 95% confidence interval [CI] −2.09, −0.63; $p$ < 0.001). There was no evidence of a difference between the "75% non-alcoholic" and the "50% non-alcoholic" (7.2%) groups (95% CI 0.10, 1.34; $p$ = 0.022) or between the "50% non-alcoholic" and the "25% non-alcoholic" groups (95% CI −1.44, 0.17; $p$ = 0.121). In the second part of a hurdle model in participants (559/607) selecting any drinks containing alcohol, the "75% non-alcoholic" group selected fewer alcohol units compared to the "50% non-alcoholic" (95% CI −0.44, −0.14; $p$ < 0.001) and "25% non-alcoholic" (95% CI

**Data Availability Statement:** All data is available on the OSF project page (https://osf.io/m85gd) and the University of Cambridge Research Repository (https://doi.org/10.17863/CAM.94093).

**Funding:** This research was funded in whole, or in part, by the Wellcome Trust [ref: 206853/Z/17/Z (TMM, GJH, MRM)]. For the purpose of Open Access, the author has applied a CC BY public copyright licence to any Author Accepted Manuscript version arising from this submission. The funders had no role in study design, data collection and analysis, decision to publish, or preparation of the manuscript.

**Competing interests:** The authors have declared that no competing interests exist.

**Abbreviations:** AUDIT, Alcohol Use Disorders Identification Test; CI, confidence interval; OSF, Open Science Framework; TIPPME, Typology of Interventions in Proximal Physical Micro-Environments.

−0.54, −0.24; $p < 0.001$) groups, with no evidence of a difference between the "50% non-alcoholic" and "25% non-alcoholic" groups (95% CI −0.24, 0.05; $p = 0.178$). Overall, across all participants, 17.46 units (95% CI 15.24, 19.68) were selected in the "75% non-alcoholic" group; 25.51 units (95% CI 22.60, 28.43) in the "50% non-alcoholic" group; and 29.40 units (95% CI 26.39, 32.42) in the "25% non-alcoholic" group. This corresponds to 8.1 fewer units (a 32% reduction) in the "75% non-alcoholic" compared to the "50% non-alcoholic" group, and 11.9 fewer alcohol units (41% reduction) compared to the "25% non-alcoholic" group; 3.9 fewer units (13% reduction) were selected in the "50% non-alcoholic" group than in the "25% non-alcoholic" group.

For all other outcomes, alcohol selection and purchasing were consistently lowest in the "75% non-alcoholic" group.

Study limitations include the setting not being entirely naturalistic due to using a simulated online supermarket as well as an actual online supermarket, and that there was substantial dropout between selection and purchasing.

## Conclusions

This study provides evidence that substantially increasing the proportion of non-alcoholic drinks—from 25% to 50% or 75%—meaningfully reduces alcohol selection and purchasing. Further studies are warranted to assess whether these effects are realised in a range of real-world settings.

## Trial registration

ISRCTN: 11004483; OSF: https://osf.io/qfupw.

---

Author summary

### Why was this study done?

- Excessive alcohol consumption contributes to the global burden of non-communicable diseases, including cancer, heart disease, and stroke. Interventions that change physical and economic environments have the potential to reduce alcohol consumption.

- Interventions targeting physical environments include availability interventions that involve changing the proportion of healthier options that are available, relative to less healthy options.

- A previous online study found that increasing the availability of non-alcoholic compared to alcoholic drinks reduced the hypothetical selection of alcoholic drinks, but there is an absence of evidence from naturalistic settings.

### What did the researchers do and find?

- This study evaluated the impact of increasing the proportion of non-alcoholic (relative to alcoholic) drinks, on selection and actual purchasing of alcohol.

- In a randomised controlled trial, 737 participants were randomly assigned to one of 3 groups with varying proportions of alcoholic versus non-alcoholic drinks ("25% non-alcoholic/75% alcoholic"; "50% non-alcoholic/50% alcoholic"; and "75% non-alcoholic/25% alcoholic").

- Participants selected drinks from 64 options in a simulated online supermarket that was designed to look and function similarly to an online supermarket. Participants were then required to immediately purchase the same drinks in an actual online supermarket.

- It was found that increasing the proportion of non-alcoholic drinks—from 25% to 50% or 75%—reduced the amount of alcohol selected and bought, in this online supermarket setting.

### What do these findings mean?

- This study provides evidence that increasing the proportion of non-alcoholic drinks could reduce alcohol selection and purchasing, highlighting the potential for availability interventions to reduce alcohol sales at the population level.

- Further studies are warranted to assess whether these effects are realised in a range of real-world settings.

## Introduction

Excessive alcohol consumption is one of 4 sets of modifiable behaviours—along with tobacco use, physical inactivity, and unhealthy diet—that make a major contribution to the global burden of non-communicable diseases, including cancer, heart disease, and stroke [1,2]. Given the influence of environmental cues upon consumption and related behaviours, interventions that change physical and economic environments in which these behaviours occur have the potential to reduce alcohol consumption [3]. Altering the availability of alcohol products has been identified as a particularly potent approach [4] but has typically been examined in relation to demographic, temporal, or spatial restrictions (e.g., by age, opening hours, or number or density of retail outlets), and not in terms of changing the range of available products. One intervention of this kind, potentially scalable to population-level and currently untested, involves increasing the proportion of non-alcoholic (relative to alcoholic) drink options that are available to select, purchase, and ultimately consume [5]. This can be achieved by either making more non-alcoholic options available, removing some alcoholic options, or by doing both and so retaining the same overall number of options [6]; the latter is assessed in the current study.

The promise of so-called "availability" interventions that change proportions of unhealthy (relative to healthier) products is highlighted by an emerging evidence base in relation to food [5]. A Cochrane systematic review found that reducing the proportion of available food products of a certain type (e.g., unhealthy snacks) resulted in markedly reduced selection of those foods [7], although the included evidence was limited in both quality and quantity. More recent field trials also suggest that decreasing the proportion of higher energy or meat-based foods reduces their consumption [8–11]. In terms of alcohol products, there is an absence of

evidence, with no eligible studies identified in the aforementioned Cochrane review [7] or in a recent search update [5]. In what is, to our knowledge, the only previous study that has examined the potential of such an availability intervention applied to alcohol, the proportion of participants selecting an alcoholic drink decreased from 74% when one-quarter of the available drinks were non-alcoholic, to 51% when three-quarters were non-alcoholic [12]. However, this study only measured hypothetical and mandatory selection of a single drink from a limited range of 8 options. Studies using meaningful outcomes and conducted within more naturalistic contexts that include wider product ranges are necessary to inform the development and implementation of real-world interventions and policies.

There is clear interest in increasing the availability of non-alcoholic drink options, from the perspective of both consumers and policymakers. While the current market for alcohol-free beer, wine, and spirits represents a 3.5% share of the global alcohol industry and is therefore relatively small, it is rapidly growing [13]. For example, low and no-alcohol beer currently accounts for 3% of the total beer market [14], but this is forecast to increase by nearly 13% per annum over the next 3 years and is the fastest growing drinks segment in the United Kingdom [15]. In 2021, the no/low alcohol market grew by 6% globally, and in the UK, sales of non-alcoholic beer increased by 7% [16].

In 2020, the UK Government made a commitment with the drinks industry to increase the availability of alcohol-free and low-alcohol products by 2025, although details on what this would involve have not been published [17]. Currently, most consumers purchase no or low alcohol drinks infrequently, although increased availability of these products is associated with an increase in their sales [18] and reductions in grams of alcohol purchased [19,20]. Non-alcoholic alternatives to alcohol (i.e., alcohol-free drinks and soft drinks marketed to adults) still only represent a small proportion of the market, however, which combined with their recent increase in popularity, suggests that there is substantial scope for increasing their availability.

The aim of the current study was to estimate the impact of increasing the proportion of non-alcoholic drink options relative to alcoholic drink options, on the number of alcohol units that are (i) selected (with the intention to purchase) and (ii) purchased. We hypothesised that increasing the availability of non-alcoholic alternatives to alcohol would reduce the number of alcohol units selected and purchased.

## Methods

The trial was prospectively registered (ISRCTN: 11004483). In addition, both the study protocol (https://osf.io/qfupw) and a detailed statistical analysis plan (https://osf.io/4yuca) were preregistered on the Open Science Framework (OSF). The study was approved by the Faculty of Life Sciences Research Ethics Committee at the University of Bristol (reference no: 116124). Trial reporting follows CONSORT 2010 guidelines.

### Study design

The study used a parallel-groups randomised controlled design. Individual participants were randomly allocated without stratification to one of 3 groups differing in the proportion (%) of non-alcoholic versus alcoholic drink options available for selection: Group 1: "25% non-alcoholic/75% alcoholic"; Group 2: "50% non-alcoholic/50% alcoholic"; Group 3: "75% non-alcoholic/25% alcoholic".

### Setting

The study was conducted online using simulated and real online supermarkets. First, participants completed a simulated supermarket selection task hosted on the Qualtrics online survey

platform (see https://osf.io/2cy7t for example task images). The simulated supermarket was designed to look and function as similarly to the actual online supermarket as was possible within Qualtrics. Drinks were displayed in rows of 4 drink options and participants could add these to their basket, which displayed a total price after the selection had been made. Following this, participants were required to purchase drinks in Tesco online supermarket (Tesco.com), the largest national supermarket in the UK.

## Participants

To be eligible for the study, participants had to be adults aged 18 years and over residing in England or Wales, who consumed beer or wine regularly (i.e., at least weekly), and purchased these drinks at least monthly from Tesco.com, with a minimum spend of £20. Participants had to be willing to complete a shop at Tesco.com following completion of the selection task, book a delivery or click-and-collect slot, and send proof of purchase (their receipt) to the research team. Similar proportions of males and females of a range of ages were recruited via Roots Research (https://rootsresearch.co.uk/), one of the largest research agencies in the UK, with a high-quality panel of over 350,000 participants. Recruitment occurred between March and July 2021.

**Sample size.**   A previous online study compared the impact on drink selection of altering non-alcoholic versus alcoholic drink availability [12]. The proportion of participants selecting an alcoholic drink decreased from 74% when non-alcoholic drink availability was low (25% of drink options), to 61% when availability was medium (50% of drink options), and 51% when availability was high (75% of drink options) (i.e., a difference of 13% and 10%, respectively, between adjacent groups). However, only a single drink was selected in this online study and there was no intention to purchase the selected drinks (i.e., decisions were purely hypothetical). As such, to our knowledge, there was no comparable evidence available from which to estimate effects of this intervention on selection or purchasing behaviour of multiple drink options. Available resources allowed recruitment of around 600 participants. As an illustrative calculation, assuming 15% attrition, a sample of 510 participants (170/group) was sufficient to detect an effect of $d = 0.3$ for the primary outcome for a two-group $t$ test with alpha of 5% and 80% power. Using pretesting data (approximately 5/group), the conservative SD estimate was 12.1 units (i.e., the maximum group variance observed), indicating that the sample size was sufficient to detect a difference of 3.7 alcohol units selected between groups.

## Randomisation and masking

Randomised assignment of participants was completed via the default algorithm in Qualtrics with a ratio of 1:1:1. Participants were unaware of their group assignment throughout the study. The research team were blinded to allocation until participants had completed the primary outcome; the statistician completing the analysis was blinded to the allocation.

## Intervention

All participants viewed a total range of 64 drink options. This comprised (i) a range of beers, ciders, alcohol-free beer and cider alternatives, and soft drinks (32 options), and (ii) a range of wines, alcohol-free wine alternatives, and soft drinks (32 options), modelled on the available range of products on Tesco.com. Initial scoping work found that Tesco.com proportions of alcoholic versus non-alcoholic options were roughly 25% non-alcoholic (360 options) and 75% alcoholic (1058 options). Alcohol-free beer, alcohol-free cider, and alcohol-free wine options used in the task were matched as far as possible on brand and size characteristics with the alcohol options available online at Tesco.com. Additional alcoholic beer, cider, and wine options were selected based on the leading brands of lager, ale, mild and stout, cider, and wine

[21–23] in Great Britain according to the number of consumers. Adult soft drinks were selected based on options that were commonly displayed next to alcoholic drinks in physical supermarkets and most likely to be consumed as a substitute for alcohol (meaning that children's soft drinks, milk, and fruit juice were excluded). Participants viewed varying proportions of non-alcoholic and alcoholic drink options depending on their assignment: Group 1: "25% non-alcoholic/75% alcoholic"; Group 2: "50% non-alcoholic/50% alcoholic"; Group 3: "75% non-alcoholic/25% alcoholic". The proportions used were based on previous food and alcohol studies [12,24]. Within each range of alcoholic drinks, there were the same number of beer as wine options, and within each range of non-alcoholic drinks, there were the same number of soft drinks as alcohol-free options. Participants were randomised to the order in which each subcategory (soft drinks; alcohol free; alcoholic) was presented within each of the beer and wine categories, and the order of drinks within each subcategory was also randomised. Each drink option was displayed as an image, below which was a text description of the drink (identical to Tesco.com), the % alcohol by volume (ABV) for drinks containing alcohol, and its price.

Full details of the task, as well as the complete list of drinks, are in the S1 Supporting information (Table A and B). In the Typology of Interventions in Proximal Physical Micro-Environments (TIPPME) [3], this is classified as an "Availability × Product" intervention, while in a detailed conceptual framework specific to availability interventions [6], this is categorised as a "Relative Availability" intervention.

## Procedure

Participants were initially provided with an information sheet, instructions, and a link to the study via email. Participants were told the study was investigating "Adult drink preferences in England and Wales" and were not made aware of the study aim. Participant instructions outlined the stages of the study in detail, i.e., that participants were required to select the drinks for their next shop from Tesco.com in a simulated online supermarket (Stage 1), then to immediately go to Tesco.com to book a delivery slot and add these drinks to their shopping basket (Stage 2), and finally to send their receipt to the study team on their delivery or collection day (Stage 3). Once they had started the study task, participants were again presented with this information and provided consent. Participants were randomised, and in a simulated online supermarket environment replicating Tesco.com (within Qualtrics), they were shown the available drink selection. They chose all the drinks they wanted to purchase in their next online shop at Tesco.com. They were then shown their total drink selection and price and given the opportunity to amend their selection before continuing. Participants then completed demographic and drinking behaviour measures.

After completing the simulated online supermarket task, participants were automatically sent an email detailing their selection. They were prompted to open this email and given further instructions for completing purchasing, alongside a direct link to Tesco.com. Participants placed their selected drinks in their Tesco.com shopping basket, along with any other items, booked their delivery or collection slot, and confirmed this within 48 hours. They were sent a reminder email on their delivery/collection day and requested to send an itemised receipt to the research team within 48 hours. Up to 2 follow-up reminders were sent, 2 and 4 days later. Purchases were recorded from receipts, including any additional drink purchases. Substitutions by the participant or by Tesco that were explained (e.g., not in stock) were marked as the original drink they attempted to purchase. Participants were debriefed via email and reimbursed £25 (approximately $35) for their time taking part in the study (but not the drinks they purchased).

## Outcome measures

**Primary outcome.** The primary outcome was the number of alcohol units selected in the context of a stated intention to purchase. In the UK, a unit is a standard measure of pure alcohol in a drink with 1 unit equivalent to 10 ml or 8 g of pure alcohol (this is equivalent to 0.56 of a US standard drink [25]). Participants were aware when selecting drinks in the task that they were required to subsequently purchase the drinks chosen and send proof of this to the research team (otherwise, they were not reimbursed). Units of alcohol were calculated for all drinks that were >0% ABV, i.e., alcoholic and "alcohol-free" drinks (which were defined as containing more than 0% and up to 0.5% ABV). This outcome was preregistered as the primary outcome as it was assessed in all participants who were exposed to the intervention and measured within the same context, i.e., the simulated online supermarket.

**Secondary and additional outcomes.** Secondary outcomes were the number of alcoholic and non-alcoholic drinks selected, the number of alcohol units purchased, and the proportion (i.e., percentage) of total drinks selected and purchased that were alcoholic. Additional outcomes were the total number of drinks selected, and purchased, the number of alcoholic drinks purchased, and the number of non-alcoholic drinks purchased.

Selection outcomes were assessed from the simulated online supermarket task, and purchasing outcomes were assessed from receipts after shops at Tesco.com were completed. Purchasing outcomes were calculated to include (i) additional drinks from study categories only (i.e., beer, cider, wine, and adult non-alcoholic drinks) and (ii) all additional drinks (i.e., all alcoholic and non-alcoholic drinks—excluding squash, juice, tea, coffee, and children's drinks).

## Other measures

**Demographic characteristics.** Age, sex, and highest qualification attained ("Higher education or professional/vocational equivalents," "A levels or vocational level 3 or equivalents," "GCSE/O level grade A*-C or vocational level 2 or equivalents," "Qualifications at level 1 and below," "Other qualifications: level unknown," or "No qualifications"). Qualifications classifications were based on UK definitions [26,27].

**Household members.** Number of adults (aged 18+) and of children (aged <18).

**Drinking behaviour risk.** Alcohol Use Disorders Identification Test (AUDIT) [28], a 10-item clinical screening measure for assessing risk associated with participants' drinking behaviour (low risk drinking: score 0 to 7; medium/hazardous risk drinking: score 8 to 15; high/harmful risk drinking: score ≥16).

**Baseline weekly unit consumption.** Self-reported drinks consumed and purchased over the previous 7 days, used to calculate the number of alcohol units as a continuous variable.

**Free-text comments.** Participants provided comments on the task, such as explaining their choice of drinks.

## Statistical analysis

Analyses were preregistered in a detailed statistical analysis plan (https://osf.io/4yuca).

All participants who completed the selection task were included in the primary outcome analysis. Participants who failed to complete the selection task and those whose responses were flagged as incomplete or suspicious—e.g., those that forged data (i.e., submitted fake receipts) or selected an unrealistically large number (with a cutoff of >100 drinks) of drinks that were not purchased—were excluded (see Fig 1 for details by group). The criteria used to exclude data were not preregistered but were defined and applied prior to data analysis, while researchers were unaware of group allocation. The data included participants that did not select any drinks, as this was still a valid choice. Participant characteristics are presented in Table 1 and

raw outcome data in Table 2. The distribution of the primary outcome was highly skewed and zero inflated, and, therefore, a hurdle model was used for analysis, fitting (i) a binary logistic model (part 1) to the zero and non-zero outcomes and (ii) a truncated negative binomial model (part 2) to just the positive values [29,30]. The model results for the positive values are therefore based only on participants who selected at least 1 drink containing alcohol (see Table C in S1 Supporting information), with non-integer variables rounded to integer values before hurdle model analysis. The marginal effect estimates, with 95% confidence interval (CI), are also presented.

For most secondary outcomes, hurdle models were repeated as per the primary outcome model. Model results for the binary outcomes (part 1 of the model) and the positive values (part 2, i.e., based on values above zero) are reported in Table 3. Marginal effect estimates for all secondary outcomes, with 95% CI, are presented in Table 4. The *p*-values for part 2 of the model and the change in marginal effect estimates (with the associated percentage reduction) are reported in the Results. For additional purchasing outcomes, negative binomial regression was required due to the skewed data. For the proportion outcomes (i.e., percentage of total drinks selected, and purchased, that were alcoholic), a beta binomial regression was used to model the proportion using the counts of relevant drinks selected out of the count of all drinks selected, and this could accommodate the bimodal distribution observed. For these outcomes only, due to the nature of the model, any participants who did not select any drink (as appropriate for the outcome) were excluded.

Two per-protocol analyses were prespecified, in which the primary outcome analysis was repeated for (i) participants who purchased what they selected, either with or without additional drinks (per-protocol analysis 1), and for (ii) participants who purchased exactly what they selected and purchased no additional drinks (per-protocol analysis 2). Model results are presented in Table 5.

For all outcomes, for the co-primary comparisons of primary interest (using the "25% non-alcoholic" group as the reference group), a 5%/2 adjustment to the interpretation threshold for statistical significance was made. For the third comparison of tertiary interest (where "75% non-alcoholic" and "50% non-alcoholic" groups were compared), a simplistic 5%/3 adjustment was made rather than using methods that may not report all *p*-values (e.g., Benjamini–Hochberg, Holm–Bonferroni). These additional tests were calculated by refitting the same model with different reference categories.

## Results

### Sample characteristics

Fig 1 shows the flow of participants. In total, 737 participants were randomised, 640 of whom completed the selection task. A total of 607 participants were included in the primary outcome analysis. The primary analysis dataset was 59.7% female and the mean age was 37.8 years (SD = 11.4; range: 18 to 76). Groups were well balanced on all characteristics (Table 1).

### Primary outcome

Raw primary outcome data are presented in Table 2, modelled estimates for each part of the hurdle model in Table 3, and the overall marginal effect estimates in Table 4.

In the first part of the hurdle model, a greater proportion of participants in the "75% non-alcoholic" group did not select any alcohol (27/206 [13.1%]) compared to the "25% non-alcoholic" group (7/207 [3.4%]; 95% CI −2.09, −0.63; *p* < 0.001); there was no evidence of a difference between the "50% non-alcoholic" (14/194 [7.2%]) and the "75% non-alcoholic" group (95% CI 0.10, 1.34; *p* = 0.022, given the adjusted threshold of *p* = 0.0167) or between the "50%

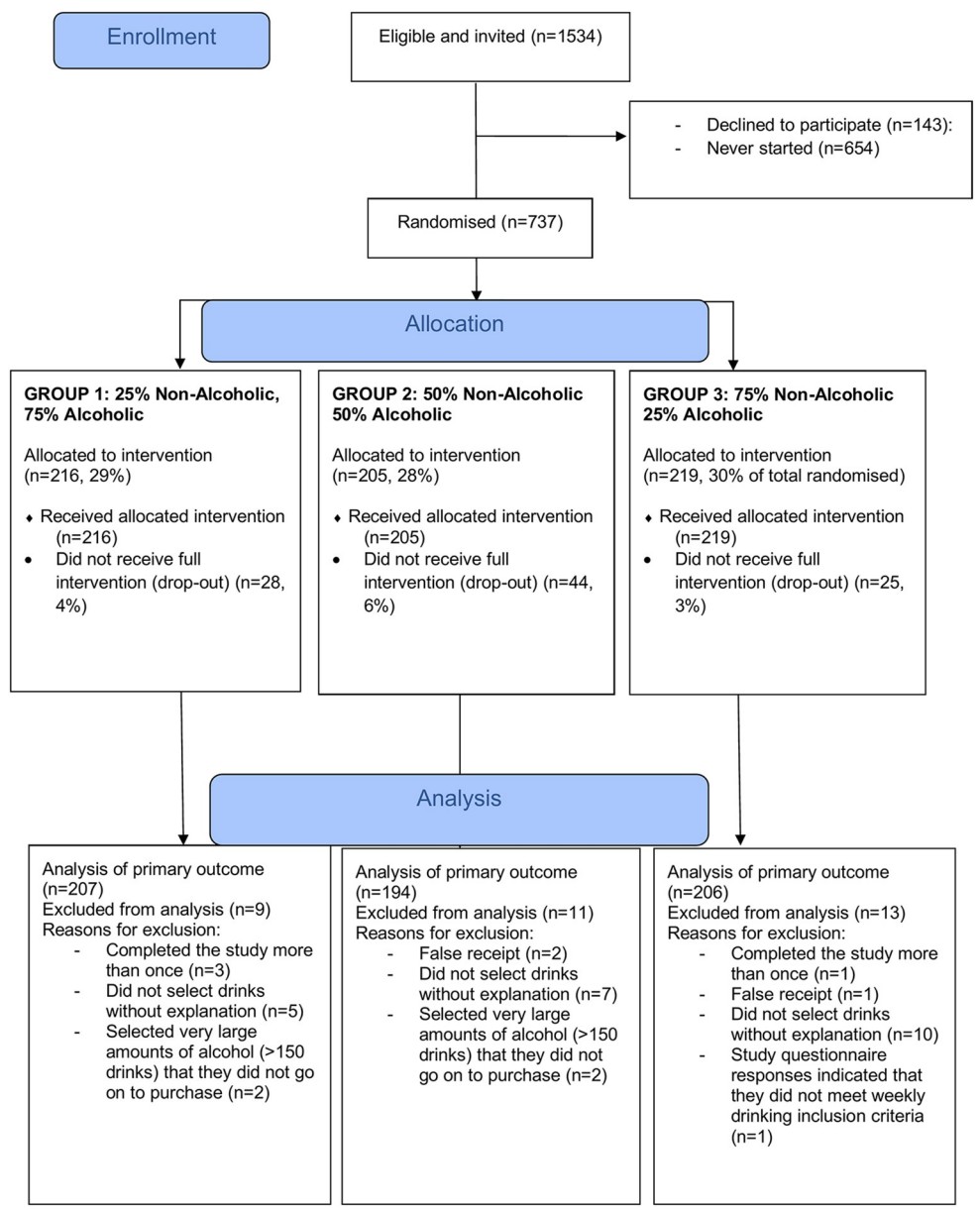

**Fig 1. Participant flowchart.**

non-alcoholic group" and the "25% non-alcoholic" group (95% CI −1.44, 0.17; $p = 0.121$). In the second part of a hurdle model in participants (559/607) selecting any drinks containing alcohol, the "75% non-alcoholic" group selected fewer alcohol units compared to the "50% non-alcoholic" (95% CI −0.44, −0.14; $p < 0.001$) and "25% non-alcoholic" (95% CI −0.54, −0.24; $p < 0.001$) groups, with no evidence of a difference between the "50% non-alcoholic" and "25% non-alcoholic" groups (95% CI −0.24, 0.05; $p = 0.178$). Overall, across all participants, 17.46 units (95% CI 15.24, 19.68) were selected in the "75% non-alcoholic" group; 25.51 units (95% CI 22.60, 28.43) in the "50% non-alcoholic" group; and 29.40 units (95% CI 26.39, 32.42) in the "25% non-alcoholic" group. This corresponds to 8.1 fewer units (32% reduction) in the "75% non-alcoholic" group compared to the "50% non-alcoholic" group, and 11.9 fewer

**Table 1. Characteristics of participants included in primary outcome analysis (n (%), unless otherwise stated).**

| | GROUP 1: 25% non-alcoholic (*n* = 207) | GROUP 2: 50% non-alcoholic (*n* = 194) | GROUP 3: 75% non-alcoholic (*n* = 206) |
|---|---|---|---|
| **Alcohol consumption previous week (units)[1a] (mean (SD))** | 25.9 (26.1) | 24.5 (22.6) | 27.7 (37.5) |
| **Alcohol purchasing previous week (units)[1b] (mean (SD))** | 41.5 (37.3) | 37.6 (28.3) | 42.2 (37.5) |
| **AUDIT score (mean (SD))[2]** | 8.8 (5.5) | 8.8 (5.4) | 8.9 (5.2) |
| - Low-risk drinking (scores 1–7) | 107 (52) | 98 (51) | 98 (48) |
| - Medium- to high-risk drinking scores (8+) | 99 (48) | 95 (49) | 106 (52) |
| **Age (mean (SD))[3]** | 37.7 (11.0) | 37.6 (11.8) | 38.1 (11.6) |
| 18–39 years | 122 (59) | 123 (63) | 132 (64) |
| 40 and over | 84 (41) | 71 (37) | 74 (36) |
| **Sex[3]** | | | |
| Male | 83 (40.3) | 82 (42.3) | 78 (37.9) |
| Female | 123 (59.7) | 112 (57.7) | 127 (61.7) |
| **Household members[3]** | | | |
| Number of adults in household (mean (SD)) | 2 (0.8) | 2.1 (1.4) | 2 (0.8) |
| Number of children in household (mean (SD)) | 1 (0.7) | 0.7 (0.9) | 0.7 (0.9) |
| **Highest qualification[3]** | | | |
| No qualifications | 2 (1.0) | 0 (0) | 1 (0.5) |
| Qualifications at level 1 and below | 0 (0) | 3 (1.5) | 0 (0) |
| GCSE/O level grade A*-C or vocational level 2 or equivalents | 23 (11.2) | 18 (9.3) | 26 (12.6) |
| A levels or vocational level 3 or equivalents | 36 (17.5) | 52 (26.8) | 42 (20.4) |
| Higher education or professional/vocational equivalents | 144 (69.9) | 121 (62.4) | 137 (66.5) |
| Other qualification | 1 (0.5) | 0 (0) | 0 (0) |

AUDIT, Alcohol Use Disorders Identification Test; SD, standard deviation.

[1]All participants in the sample explicitly reported drinking at least once a week in the screener questions. A further weekly drinking measure recorded the amount of alcohol consumed (1a) and purchased (1b) in the previous week as an overall indication of the volume of alcohol consumed and purchased weekly. UK definition of alcohol units is used: In the US, this is equivalent to 0.564 standard drinks [25].

[2]Heavy and binge drinking behaviours (AUDIT), scores 1–7 indicative of low-risk drinking; 8–14: hazardous alcohol consumption; 15 +: moderate–severe alcohol use. Missing data for 3 participants.

[3]Missing data for 1 participant.

alcohol units compared to the "25% non-alcoholic" group (a 41% reduction); 3.9 fewer units (13% reduction) were selected in the "50% non-alcoholic" group than in the "25% non-alcoholic" group.

## Secondary outcomes

Raw secondary outcome data are presented in Table 2, modelled estimates for each part of the hurdle model in Table 3, and the overall marginal effect estimates in Table 4. For purchasing outcomes, of the 640 participants who completed the selection task, 422 (66%) went on to purchase drinks from Tesco.com. Attrition from selection to purchasing stages was very similar across the 3 randomised groups (with 136, 141, and 145 completing purchasing).

Results for all secondary selection and purchasing outcomes demonstrated a wholly consistent pattern of results with amounts and proportions of alcohol selected and purchased consistently lowest in the "75% non-alcoholic" group, although not always significantly so.

**Table 2. Primary and secondary outcomes (raw means (SD)).**

|  | GROUP 1: 25% non-alcoholic (*n* = 207) | GROUP 2: 50% non-alcoholic (*n* = 194) | GROUP 3: 75% non-alcoholic (*n* = 206) |
|---|---|---|---|
|  | Mean (SD) | | |
| **Primary outcome**: Number of alcohol units[1] selected (with an intention to purchase). | 29.5 (29.8) | 25.6 (20.5) | 17.6 (16.2) |
| **Secondary outcomes: selection** | | | |
| Number of alcoholic drinks selected | 10.6 (14.0) | 8.8 (9.2) | 6.4 (7.1) |
| Number of non-alcoholic drinks selected | 5.4 (13.6) | 6.4 (10.5) | 8.8 (15.0) |
| Percentage of total drinks selected that are alcoholic | 75% (34%) | 64% (34%) | 52% (37%) |
| **Secondary outcomes: purchasing** | | | |
|  | GROUP 1: 25% non-alcoholic (*n* = 145) | GROUP 2: 50% non-alcoholic (*n* = 141) | GROUP 3: 75% non-alcoholic (*n* = 136) |
| Number of alcohol units purchased (including additional drinks from study categories only) | 26.7 (18.6) | 28.7 (23.3) | 23.4 (30.4) |
| Number of alcohol units purchased (including all additional drinks) | 29.1 (22.5) | 30.7 (26.9) | 28.7 (36.6) |
| Percentage of total drinks purchased that are alcoholic (including additional drinks from study categories only) | 76% (34%) | 68% (32%) | 55% (37%) |
| Percentage of total drinks purchased that are alcoholic (including all additional drinks) | 68% (36%) | 61% (33%) | 52% (36%) |

[1]N.B. UK definition of alcohol units is used: In the US, this is equivalent to 0.564 standard drinks [25].

**Selection.** Participants in the "75% non-alcoholic" group selected fewer alcoholic drinks than those in the "25% non-alcoholic" group (95% CI −0.66, −0.20; *p* < 0.001), with an overall difference between marginal effect estimates of 4.1 drinks, equivalent to a 43% reduction. There were non-significant reductions in the "75% non-alcoholic" compared to the "50% non-alcoholic" group (95% CI −0.50, −0.02; *p* = 0.03; overall difference between marginal effect estimates of 2.3 drinks, 30% reduction) and in the "50% non-alcoholic" group compared to the "25% non-alcoholic" group (95% CI −0.39, 0.06; *p* = 0.148; overall difference between marginal effect estimates of 1.8 drinks, 19% reduction).

There was no evidence of a difference in the number of non-alcoholic drinks selected between groups.

The percentage of total drinks selected that were alcoholic was lower in the "75% non-alcoholic" group (52%, 95% CI 47%, 57%) compared to the "50% non-alcoholic" group (65%, 95% CI 60%, 70%; *p* < 0.001), and lower compared to the "25% non-alcoholic" group (78%, 95% CI 74%, 82%; *p* < 0.001); the percentage of total drinks selected that were alcoholic was also lower in the "50% non-alcoholic" group compared to the "25% non-alcoholic" group (*p* < 0.001).

**Purchasing.** When including additional drinks that were purchased from study categories only, there was a reduction in alcohol units purchased in the "75% non-alcoholic" group compared to the "50% non-alcoholic" group (95% CI −0.42, −0.09; *p* = 0.003, overall difference between marginal effect estimates of 7.3 units, 26% reduction). There was a non-significant reduction in the "75% non-alcoholic" compared to the "25% non-alcoholic" group (95% CI −0.32, −0.00; *p* = 0.056), with an overall difference between marginal effect estimates of 5.3 units, equivalent to a 20% reduction. There was no evidence of a difference in alcohol units purchased between the "50% non-alcoholic" and the "25% non-alcoholic" groups. There was evidence that the percentage of total drinks selected that were alcoholic was lower in the "75% non-alcoholic" group (55%, 95% CI 49%, 61%) compared to the "50% non-alcoholic" group (67%, 95% CI 61%, 72%; *p* = 0.004) and to the "25% non-alcoholic" group (78%, 95% CI 73%,

**Table 3. Model results for primary and secondary outcomes: Estimates from hurdle models (95% confidence intervals), p-values.**

| | | Compared to reference group: 25% non-alcoholic (n = 207) | | Compared to reference group: 50% non-alcoholic (n = 194)[1] |
|---|---|---|---|---|
| | | **50% non-alcoholic (n = 194)** | **75% non-alcoholic (n = 206)** | **75% non-alcoholic (n = 206)** |
| **Primary outcome**: Number of alcohol units[2] selected (with an intention to purchase) | Hurdle model part 1: binary outcomes | −0.64 (95% CI −1.44, 0.17) $p = 0.121$ | −1.36 (95% CI −2.09, −0.63) $p < 0.001$ | 0.72 (95% CI 0.10, 1.34) $p = 0.022$ |
| | Hurdle model part 2: non-zero outcomes | −0.10 (95% CI −0.24, 0.05) $p = 0.178$ | −0.39 (95% CI −0.54, −0.24) $p < 0.001$ | −0.29 (95% CI −0.44, −0.14) $p < 0.001$ |
| **Secondary outcomes: selection** | | | | |
| Number of alcoholic drinks selected[3] | Hurdle model part 1: binary outcomes | −0.51 (95% CI −1.27, 0.25) $p = 0.189$ | −1.27 (95% CI −1.95, −0.59) $p < 0.001$ | −0.76 (95% CI −1.36, −0.16) $p = 0.013$ |
| | Hurdle model part 2: non-zero outcomes | −0.17 (95% CI −0.39, 0.06) $p = 0.148$ | −0.43 (95% CI −0.66, −0.20) $p < 0.001$ | −0.26 (95% CI −0.50, −0.02) $p = 0.03$ |
| Number of non-alcoholic drinks selected[3] | Hurdle model part 1: binary outcomes | 0.85 (95% CI 0.45, 1.25) $p < 0.001$ | 1.30 (95% CI 0.89, 1.72) $p < 0.001$ | 0.46 (95% CI 0.03, 0.88) $p = 0.034$ |
| | Hurdle model part 2: non-zero outcomes | −0.27 (95% CI −0.63, 0.10) $p = 0.148$ | −0.06 (95% CI −0.41, 0.29) $p = 0.735$ | 0.21 (95% CI −0.11, 0.53) $p = 0.197$ |
| Percentage of total drinks selected that are alcoholic[4] | Beta-binomial regression | −0.63, $p < 0.001$ | −1.27, $p < 0.001$ | −0.54, $p < 0.001$ |
| **Secondary outcomes: purchasing** | | | | |
| | | Compared to reference group: 25% non-alcoholic (n = 145) | | Compared to reference group: 50% non-alcoholic (n = 141) |
| | | **50% non-alcoholic (n = 141)** | **75% non-alcoholic (n = 136)** | **75% non-alcoholic (n = 136)** |
| Number of alcohol units purchased (including additional drinks from study categories only)[3] | Hurdle model part 1: binary outcomes | −0.27 (95% CI −1.22, 0.69) $p = 0.584$ | −0.76 (95% CI −1.65, 0.13) $p = 0.100$ | −0.49 (95% CI −1.33, 0.34) $p = 0.248$ |
| | Hurdle model part 2: non-zero outcomes | 0.09 (95% CI −0.07, 0.25) $p = 0.263$ | −0.16 (95% CI −0.32, 0.00) $p = 0.056$ | −0.25 (95% CI −0.42, −0.09) $p = 0.003$ |
| Number of alcohol units purchased (including all additional drinks)[3] | Hurdle model part 1: binary outcomes | 0.85 (95% CI 0.45, 1.25) $p < 0.001$ | 1.30 (95% CI 0.89, 1.72) $p < 0.001$ | 0.46 (95% CI 0.03, 0.88) $p = 0.034$ |
| | Hurdle model part 2: non-zero outcomes | 0.06 (95% CI −0.11, 0.24) $p = 0.471$ | −0.04 (95% CI −0.22, 0.14) $p = 0.658$ | −0.10 (95% CI −0.28, 0.07) $p = 0.255$ |
| Percentage of total drinks purchased that are alcoholic (including additional drinks from study categories only)[4] | Beta-binomial regression | −0.57, $p = 0.003$ | −1.09, $p < 0.001$ | −0.51, $p = 0.004$ |
| Percentage of total drinks purchased that are alcoholic (including all additional drinks)[4] | Beta-binomial regression | −0.42, $p = 0.015$ | −0.76, $p < 0.001$ | −0.33, $p = 0.049$ |

[1]Significance threshold is 0.0167 for a 5% alpha.

[2]UK definition of alcohol units is used: In the US, this is equivalent to 0.564 standard drinks.

[3]Part 2 of the model is a negative binomial regression; therefore, the back-transformed 95% confidence intervals become asymmetric. *P* values for hurdle models are based on z-statistics from a hurdle model fitted from the normal distribution using the glmmTMB routine in R [29].

[4]Beta-binomial regression models used for analysis and *p*-values were calculated using the "oad" R package for the analysis of overdispersed data [48,49].

83%; $p < 0.001$); the percentage of total drinks selected that were alcoholic was also lower in the "50% non-alcoholic" group compared to the "25% non-alcoholic" group ($p = 0.003$).

When including all additional drinks from any category, there was no evidence of a difference between any of the groups for alcohol units purchased. There was evidence of a difference in the percentage of total drinks selected that were alcoholic between the "75% non-alcoholic" group (52%, 95% CI 46%, 58%) and the "25% non-alcoholic" group (70%, 95% CI 64%, 75%; $p < 0.001$), and the "50% non-alcoholic" group and the "25% non-alcoholic" group (60%, 95%

**Table 4. Marginal effect estimates predicted from hurdle models (95% confidence intervals) for primary and secondary outcomes.**

| | GROUP 1: 25% non-alcoholic (*n* = 207) | GROUP 2: 50% non-alcoholic (*n* = 194) | GROUP 3: 75% non-alcoholic (*n* = 206) |
|---|---|---|---|
| **Primary outcome**: Number of alcohol units selected (with an intention to purchase). | 29.40 (95% CI 26.39, 32.42) | 25.51 (95% CI 22.60, 28.43) | 17.46 (95% CI 15.24, 19.68) |
| Number of alcoholic drinks selected | 9.50 (95% CI 7.88, 11.11) | 7.74 (95% CI 6.46, 9.02) | 5.40 (95% CI 4.41, 6.40) |
| Number of non-alcoholic drinks selected | 4.98 (95% CI 3.60, 6.35) | 4.98 (95% CI 3.60, 6.35) | 7.10 (95% CI 5.36, 8.83) |
| Percentage of total drinks selected that are alcoholic | 78% (95% CI 74%, 82%) | 65% (95% CI 60%, 70%) | 52% (95% CI 47%, 57%) |
| **Secondary outcomes: purchasing** | | | |
| | GROUP 1: 25% non-alcoholic (*n* = 145) | GROUP 2: 50% non-alcoholic (*n* = 141) | GROUP 3: 75% non-alcoholic (*n* = 136) |
| Number of alcohol units purchased (including additional drinks from study categories only) | 26.66 (95% CI 23.50, 29.81) | 28.70 (95% CI 25.17, 32.24) | 21.38 (95% CI 18.49, 24.28) |
| Number of alcohol units purchased (including all additional drinks) | 28.99 (95% CI 25.37, 32.61) | 30.61 (95% CI 26.57, 34.65) | 26.64 (95% CI 23.04, 30.24) |
| Percentage of total drinks purchased that are alcoholic (including additional drinks from study categories only) | 78% (95% CI 73%, 83%) | 67% (95% CI 61%, 72%) | 55% (95% CI 49%, 61%) |
| Percentage of total drinks purchased that are alcoholic (including all additional drinks) | 70% (95% CI 64%, 75%) | 60% (95% CI 54%, 65%) | 52% (95% CI 46%, 58%) |
| **Per-protocol analyses** | | | |
| | GROUP 1: 25% non-alcoholic | GROUP 2: 50% non-alcoholic | GROUP 3: 75% non-alcoholic |
| Per-protocol analysis 1: number of alcohol units selected (*n* = 344) | 25.11 (95% CI 21.98, 28.24) | 23.58 (95% CI 20.54, 26.62) | 16.68 (95% CI 14.22, 19.14) |
| Per-protocol analysis 1: number of alcohol units purchased (including additional drinks from study categories only) (*n* = 343) | 26.59 (95% CI 23.22, 29.97) | 28.40 (95% CI 24.55, 32.25) | 19.95 (95% CI 17.00, 22.91) |
| Per-protocol analysis 1: number of alcohol units purchased (including all additional drinks) (*n* = 343) | 28.53 (95% CI 24.44, 32.62) | 30.25 (95% CI 25.97, 34.52) | 25.17 (95% CI 21.09, 29.25) |
| Per-protocol analysis 2: number of alcohol units selected (maps directly onto purchasing) (*n* = 182) | 23.96 (95% CI 20.01, 27.91) | 22.59 (95% CI 18.77, 26.41) | 16.42 (95% CI 12.98, 19.85) |

Marginal effect estimates were calculated using the package "Effect Displays in R for Generalised Linear Models" [50].

CI 54%, 65%; *p* = 0.015); there was no evidence of a difference between the "75% non-alcoholic" and the "50% non-alcoholic" groups.

## Per-protocol analyses

Of the 422 participants who purchased drinks, 344 participants purchased all the drinks they had selected in the selection task and 78 participants had one or more missing drinks. Of the 344 participants that purchased all the drinks they selected, 182 purchased no additional drinks.

Chi-squared tests indicated that there was no evidence against assuming equal attrition occurred. Exploratory analyses indicated that attrition was greater among participants with higher baseline alcohol purchasing, but regression using an interaction term suggested this did not bias the comparisons between groups, as there was no evidence of an effect at the usual threshold for interaction terms (*p* = 0.01) (Table D and E in S1 Supporting information). See Table 5 for model results.

In participants (*n* = 344) who completed purchasing of the drinks they had selected, either with or without additional drinks, those assigned to the "75% non-alcoholic" group selected fewer alcohol units than the "50% non-alcoholic" group (95% CI −0.48, −0.14; *p* < 0.001, overall difference between marginal effect estimates of 6.9 units, 29% reduction); the "75% non-alcoholic" group also selected fewer alcohol units than those in the "25% non-alcoholic" group (95% CI −0.50, −0.16; *p* < 0.001, overall difference between marginal effect estimates of 8.4

**Table 5. Per-protocol analyses for participants that purchased drinks: model estimates (95% confidence intervals), *p*-values[1].**

| | | Reference group: 25% non-alcoholic | | Reference group: 50% non-alcoholic |
| --- | --- | --- | --- | --- |
| | | 50% non-alcoholic | 75% non-alcoholic | 75% non-alcoholic[2] |
| Per-protocol analysis 1: number of alcohol units selected (*n* = 344) | Hurdle model part 1: binary outcomes | −0.59 (95% CI −1.64, 0.45) *p* = 0.267 | −0.95 (95% CI −1.95, 0.06) *p* = 0.065 | −0.35 (95% CI −1.22, 0.52) *p* = 0.428 |
| | Hurdle model part 2: non-zero outcomes | −0.02 (95% CI −0.19, 0.14) *p* = 0.780 | −0.33 (95% CI −0.50, −0.16) *p* < 0.001 | −0.31 (95% CI −0.48, −0.14) *p* < 0.001 |
| Per-protocol analysis 1: number of alcohol units purchased (including additional drinks from study categories only) (*n* = 343) | Hurdle model part 1: binary outcomes | −0.31 (95% CI −1.34, 0.71) *p* = 0.546 | −0.50 (95% CI −1.50, 0.50) *p* = 0.329 | −0.12 (95% CI −1.12, 0.76) *p* = 0.702 |
| | Hurdle model part 2: non-zero outcomes | 0.09 (95% CI −0.09, 0.26) *p* = 0.328 | −0.25 (95% CI −0.43, −0.07) *p* = 0.006 | −0.34 (95% CI −0.52, −0.16) *p* < 0.001 |
| Per-protocol analysis 1: number of alcohol units purchased (including all additional drinks) (*n* = 343) | Hurdle model part 1: binary outcomes | −0.32 (95% CI −1.34, 0.71) *p* = 0.546 | −0.38 (95% CI −1.51, 0.64) *p* = 0.463 | −0.07 (95% CI −1.03, 0.90) *p* = 0.889 |
| | Hurdle model part 2: non-zero outcomes | 0.08 (95% CI −0.11, 0.27) *p* = 0.414 | −0.10 (95% CI −0.29, 0.10) *p* = 0.324 | −0.18 (95% CI −0.38, 0.02) *p* = 0.078 |
| Per-protocol analysis 2: number of alcohol units selected (maps directly onto purchasing) (*n* = 182) | Hurdle model part 1: binary outcomes | −0.32 (95% CI −1.69, 1.04) *p* = 0.642 | −0.87 (95% CI −2.16, 0.41) *p* = 0.183 | −0.55 (95% CI −1.76, 0.66) *p* = 0.373 |
| | Hurdle model part 2: non-zero outcomes | −0.04 (95% CI −0.25, 0.17) *p* = 0.689 | −0.30 (95% CI −0.53, −0.08) *p* = 0.009 | −0.26 (95% CI −0.49, −0.03) *p* = 0.028 |

[1]Part 2 of the model is a negative binomial regression; therefore, the back-transformed 95% confidence intervals become asymmetric. *P* values for hurdle models are based on z-statistics from a hurdle model fitted from the normal distribution using the glmmTMB routine in R [29].

[2]Note significance threshold is 0.0167 for a 5% alpha.

units, equivalent to a 34% reduction). There was no evidence of a difference between the "50% non-alcoholic" and the "25% non-alcoholic" groups. For purchasing, when including additional drinks from study categories only, fewer alcohol units were purchased in the "75% non-alcoholic" group compared to the "50% non-alcoholic" group (95% CI −0.52, −0.16; *p* < 0.001, overall difference between marginal effect estimates of 8.5 units, 30% reduction), and in the "75% non-alcoholic" group compared to the "25% non-alcoholic" group (95% CI −0.43, −0.07; *p* = 0.006, overall difference between marginal effect estimates of 6.6 units, 25% reduction). There was no evidence of a difference between the "25% non-alcoholic" and "50% non-alcoholic" groups, and no evidence of a difference between groups for purchasing when including all additional drinks.

In participants (*n* = 182) who completed purchasing of only the drinks they had selected with no additional drinks, those assigned to the "75% non-alcoholic" group selected and purchased fewer alcohol units than did those in the "25% non-alcoholic" group (95% CI −0.53, −0.08; *p* = 0.009, overall difference between marginal effect estimates of 7.5 units, 31% reduction). There was no evidence of a difference for the other comparisons.

Full results for the additional outcomes can be found in S1 Supporting information (Table F).

## Discussion

Our data show that substantially increasing the proportion of non-alcoholic drinks relative to alcoholic drinks meaningfully reduced the amount of alcohol selected and purchased in an online supermarket context. Compared to when the majority of options were alcoholic, participants selected 41% fewer alcohol units when the majority of options were non-alcoholic, and 32% fewer alcohol units when half the options were non-alcoholic. Participants also went on to purchase significantly fewer alcohol units when the majority of options were non-alcoholic.

Importantly, the overall pattern of results was consistent for all outcomes, with amounts and proportions of alcohol selected and purchased always lowest when non-alcoholic drinks were most available, including for prespecified per-protocol analyses.

The findings of the current study are consistent with a single prior study that found increasing the proportion of non-alcoholic drinks options in an online setting reduced hypothetical selection of alcohol [12]. More generally, they are consistent with a growing body of studies that apply similar availability interventions to food [7,8,10], suggesting that these interventions have the potential to be usefully applied across different product contexts [5].

## Strengths and limitations

To our knowledge, this study is the first randomised controlled trial using a naturalistic setting to estimate the impact of increasing the proportion of non-alcoholic drinks. Meaningful selection and actual purchasing outcomes were assessed, with participants able to complete their typical online shop, including selecting and purchasing multiple options from a wide range of drinks.

The study had some limitations. First, while the primary selection outcome was assessed in the context of intention to subsequently purchase and was minimally affected by attrition, there was substantial dropout between selection and actual purchasing outcomes. However, attrition between groups was very similar by study condition, and there was sufficient power to detect effects despite this; as there is an absence of studies that look at purchasing of alcohol in this setting, effect sizes could not be anticipated, but large effects on purchasing were observed. While substantial attrition is expected in studies of this nature because of time between selection and purchasing, more generally, it may be hard to avoid for any measure of unconstrained purchasing in a real-world online supermarket. Although we are not aware of other directly comparable studies in this context, more generally, "cart abandonment"—where people do not purchase items they put in their shopping cart—is common in online (including supermarket) shopping contexts [31]. Future studies may be able address this through more intensive initial screening or follow-up of participants, or by forcing participants to immediately complete their online shop. However, such processes would arguably be less naturalistic, and including only the most motivated participants risks including a less representative sample.

Second, although the setting was as naturalistic as was feasible and actual purchasing outcomes were measured, the process involved 2 stages. Drinks were initially selected within a simulated online supermarket, before purchasing was completed in an actual online supermarket (albeit with the visual presentation of the former modelled on the latter). The principal purpose of including a measure of purchasing in the actual online supermarket was to validate and strengthen our primary outcome of selection, rather than to measure purchasing behaviour in a separate context. However, this meant that additional drink options were available in the real online supermarket, and participants could not be prevented from buying these if they wished to. As a result, the clearest effects on purchasing behaviour were in participants that followed the protocol as instructed and only purchased what they selected in the simulated supermarket where the intervention was implemented. To avoid this, the intervention would ideally have been implemented entirely within a real online supermarket. However, to our knowledge, this is the first study of an availability intervention to make use of such a setting (albeit in conjunction with a simulated supermarket component), although simulated retail settings, both online and physical, are commonly used in similar intervention studies [32–36]. This represents the most robust design used to date and could provide a useful method through which to assess interventions without requiring complex collaboration with commercial retailers,

although further research is needed to assess its external validity. Finally, while participants were largely representative of Tesco.com shoppers [37], they were mostly of higher socioeconomic position. The generalisability of these findings to disadvantaged populations therefore needs consideration, particularly as buying alcohol-free drinks is more likely to occur in less socially and materially deprived households [18].

## Implications for research and policy

This study suggests that increasing the available non-alcoholic options and reducing the available alcoholic options has the potential to meaningfully reduce selection and purchasing of alcohol. Although there was some evidence of a reduction in alcohol selected and purchased when half of the options available were non-alcoholic, effects were only consistently observed when non-alcoholic drinks became the majority. Currently, supermarkets typically stock a wider range of alcoholic than non-alcoholic alternatives to alcohol, and these results suggest that if non-alcoholic options were to become the majority instead, we might expect to see substantial reductions in alcohol purchasing. As it is yet to be seen if such major changes in ranges of drinks are feasible in real-world settings, these findings are most reasonably interpreted as proof of principle, rather than able to directly inform policy options. It is plausible that this situation could rapidly change, however. The increase in popularity of alcohol-free drinks is relatively recent, with the global market growing substantially in the last 4 years, and in the UK, it is forecast to continue to increase [15]. This recent increase in the popularity of alcohol-free drinks has led to the emergence of drinking settings reflecting this, such as an alcohol-free off licence in London [38]. In food retail contexts, there have been substantial changes seen in healthier or more sustainable ranges—such as the introduction of 50% plant-based menus [39] and the requirement to provide at least 50% healthier options in healthcare settings in Scotland [40]—suggesting that shifts of such magnitude are possible. However, before policy recommendations are made, a robust evidence base suggesting potential effectiveness is required [5], which this study provides an initial step towards. Future studies should investigate the impact of smaller and more granular alterations in proportions of non-alcoholic drinks, and in a wider range of field settings, to establish how such interventions could be used. Given the relatively large effects observed in this study, subtler interventions could elicit smaller effects that would nonetheless remain meaningful for population health, especially when considering the inherent potential for scalability across retail settings.

This intervention simultaneously increased the number of non-alcoholic drinks and decreased the number of alcoholic drinks while the overall number of drinks remained constant. It is unclear whether the effect is predominantly driven by one or the combination of these changes. Further studies are needed to disentangle this and investigate potential mechanisms more broadly, noting that there is some preliminary exploration of possible mechanisms in food contexts [6,41,42]. Importantly, the overall number of drinks that participants selected and purchased remained similar between groups, suggesting that effects were a result of shifting, rather than necessarily restricting, choices. This implies overall drink sales and, potentially, revenues may be relatively unchanged if such an intervention were to be implemented, albeit dependent on non-alcoholic drink pricing. Increasing non-alcoholic drink availability could also ultimately lead to a greater range of alcohol-free drinks and soft drinks being manufactured, further increasing their popularity in synergistic fashion [18], and many alcohol companies have already committed to this [17]. It is important to note that many alcohol-free alternatives are marketed by the alcohol industry and there is no regulation on the often-exaggerated health claims that are made about these drinks [43]. Such industry involvement has potential harms and should be monitored closely [44–47]. In addition, although some of the

non-alcoholic drink options in the current study contained no sugar and were generally lower in calories than the alcoholic options (an average of 64 calories per non-alcoholic drink versus 233 calories per alcoholic drink), many soft drinks and alcohol-free alternatives still contain large amounts of sugar and calories. Given the health risks associated with sugary drink consumption [47], continued regulation and policies to reduce sugar content and consumption from both alcoholic and non-alcoholic drinks is needed to mitigate these risks.

## Conclusions

This randomised controlled trial is the first to date—to our knowledge—to assess the effect on selection and purchasing of increasing the proportion of non-alcoholic drinks available. The findings provide evidence that substantially increasing the proportion of non-alcoholic drinks —from 25% to 50% or 75%—meaningfully reduces alcohol selection and purchasing in an online supermarket context. While these findings highlight the potential for reducing alcohol sales at the population level, further studies are warranted to assess whether these effects are realised in a range of real-world settings.

## Supporting information

**S1 Supporting information.** Table A. Proportions of non-alcoholic and alcoholic drinks displayed in the selection task. Table B. Drink options used in the selection task. Table C. Number of zero values in the selection task by group and outcome. Table D. Effect of attrition from selection to purchasing on groups. Table E. Effect of attrition from selection to purchasing on weekly units purchased at baseline. Table F. Additional outcomes: Raw means. Table G. Full model results for additional outcomes.
(DOCX)

**S1 Study protocol. Preregistered study protocol.**
(DOCX)

**S1 Analysis plan. Preregistered analysis plan.**
(DOCX)

**S1 CONSORT checklist. Checklist of information to include when reporting a randomised controlled trial (RCT).** CONSORT, Consolidated Standards of Reporting Trials.
(DOC)

## Author Contributions

**Conceptualization:** Natasha Clarke, Anna K. M. Blackwell, Marcus R. Munafò, Theresa M. Marteau, Gareth J. Hollands.

**Data curation:** Natasha Clarke, Katie De-Loyde, Mark A. Pilling.

**Formal analysis:** Katie De-Loyde, Mark A. Pilling.

**Funding acquisition:** Marcus R. Munafò, Theresa M. Marteau, Gareth J. Hollands.

**Methodology:** Anna K. M. Blackwell, Theresa M. Marteau, Gareth J. Hollands.

**Project administration:** Natasha Clarke, Anna K. M. Blackwell, Jennifer Ferrar.

**Supervision:** Gareth J. Hollands.

**Writing – original draft:** Natasha Clarke, Mark A. Pilling, Gareth J. Hollands.

**Writing – review & editing:** Natasha Clarke, Anna K. M. Blackwell, Jennifer Ferrar, Katie De-Loyde, Marcus R. Munafò, Theresa M. Marteau, Gareth J. Hollands.

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
