## [Editor Report · Decision Letter 0]

11 May 2022

Dear Dr Clarke, 

Thank you for submitting your manuscript entitled "Impact on alcohol selection and purchasing of increasing the proportion of non-alcoholic versus alcoholic drinks: randomised controlled trial" for consideration by PLOS Medicine.

Your manuscript has now been evaluated by the PLOS Medicine editorial staff and I am writing to let you know that we would like to send your submission for external assessment.

However, we first need you to complete your submission by providing the metadata that are required for full assessment. To this end, please login to Editorial Manager where you will find the paper in the 'Submissions Needing Revisions' folder on your homepage. Please click 'Revise Submission' from the Action Links and complete all additional questions in the submission questionnaire.

Please re-submit your manuscript within two working days, i.e. by May 13 2022 11:59PM.

Once your full submission is complete, your paper will undergo a series of checks in preparation for full assessment. 

Kind regards,

Richard Turner, PhD

rturner@plos.org

---

## [Decision Letter · Decision Letter 1]

12 Jun 2022

Dear Dr. Clarke,

Thank you very much for submitting your manuscript "Impact on alcohol selection and purchasing of increasing the proportion of non-alcoholic versus alcoholic drinks: randomised controlled trial" (PMEDICINE-D-22-01553R1) for consideration at PLOS Medicine. 

Your paper was discussed with an academic editor with relevant expertise and sent to independent reviewers, including a statistical reviewer. The reviews are appended at the bottom of this email and any accompanying reviewer attachments can be seen via the link below:

[LINK]

In light of these reviews, we will not be able to accept the manuscript for publication in the journal in its current form, but we would like to invite you to submit a revised version that addresses the reviewers' and editors' comments fully. You will appreciate that we cannot make a decision about publication until we have seen the revised manuscript and your response, and we expect to seek re-review by one or more of the reviewers. 

We hope to receive your revised manuscript by Jul 04 2022 11:59PM. Please email us (plosmedicine@plos.org) if you have any questions or concerns.

Please let me know if you have any questions, and we look forward to receiving your revised manuscript. 

Sincerely,

Richard Turner, PhD

Senior editor, PLOS Medicine

rturner@plos.org

Our academic editor suggested adopting somewhat more cautious language given the research design

We suggest the following amendment so that the title better matches journal style: "Alcohol selection and online purchasing with altered proportions of available non-alcoholic versus alcoholic drinks: A randomised controlled trial".

Please combine the "Methods" and "Results" subsections of the abstract. 

Please add a new final sentence to the new combined subsection, beginning "Study limitations include ..." and quoting 2-3 of the study's main limitations. 

Please quote the study dates in the abstract, and mention the country in which the study was done. 

Please also quote aggregate demographic details for study participants. 

In the abstract and throughout the text, please quote p values alongside 95% CI, where available. 

After the abstract, please add a new and accessible "Author summary" section in non-identical prose. You may find it helpful to consult one or two recent research papers in PLOS Medicine to get a sense of the preferred style. 

Please amend the heading of fig 1 to "Participant flowchart" or similar. 

Please convert "18+", for example, to "aged 18 years and over", or similar. 

Please substitute "sex" for "gender" where appropriate. 

Please restructure the early part of the Discussion section (main text) so that the first paragraph is limited to summarizing the study's findings. 

Please avoid "the first" and similar claims, and where unavoidable add "to our knowledge" or similar. 

Throughout the text, please amend reference call-outs to the following style: "... interventions to food [7,8,10], ..." (noting the absence of spaces within the square brackets). 

In the reference list, please convert all italics to plain text. 

Where appropriate, please list 6 author names rather than 3, followed by 'et al.'.

Please remove all iterations of "[Internet]" from the reference list.

Please remove the information on data access, competing interests and funding from the end of the main text. In the event of publication, this information will appear in the article metadata via entries in the submission form, and does not need to be repeated here. 

Please include a completed CONSORT checklist with your revision, labelled "S1_CONSORT_Checklist" or similar and referred to as such in the Methods section (main text). 

In the checklist, please refer to individual items by section (e.g., "Methods") and paragraph numbers, not by line or page numbers as these generally change in the event of publication. 

Comments from the reviewers:

*** Reviewer #1: 

Thank you for the opportunity to review PMEDICINE 22-01553R1, Impact on alcohol selection and purchasing of increasing the proportion of non-alcoholic versus alcoholic drinks: randomized controlled trial. This study used a naturalistic randomized controlled trial to examine the impact of availability of alcoholic drinks and non-alcoholic drinks on consumers' alcohol selections. The authors report that increasing the proportion of drinks to be 75% non-alcoholic/25% alcoholic reduces alcohol selections compared to a 50%/50% and 25%/75% non-alcoholic/alcoholic product mix. The topic of the paper - strategies to reduce alcohol consumption - is highly important and timely given the growing burden of alcohol-related morbidity and mortality globally. The intervention tested is interesting, though its policy-relevance is not clear. Major strengths of the paper include the randomized design and the naturalistic online shopping task paired with actual shopping in an online supermarket. I had concerns with the statistical analysis and potential policy relevance of the study, along with several more minor comments.

Major Concerns 

1. The authors note that the primary outcome (and a number of the secondary outcomes) was highly skewed and zero inflated, and therefore chose to fit a hurdle model. Use of a hurdle model is appropriate; however, I was surprised to see that the authors state that they report in the Results "model results for positive values." This means that the primary results reported in the text exclude a substantial number of participants (those who didn't select any alcohol) and do not incorporate any potential effect of the interventions on probability of purchasing any alcoholic drinks. To my knowledge, this is not the standard way to report two-part models such as hurdle models. More typically, results from the two parts of the model are combined and a single marginal effect is reported (this can be accomplished using built-in commands in a variety of statistical packages or using bootstrapping). At the least, it is also not appropriate to focus the abstract and main text results on just one part of the model and largely leave the results from the first part to the supplement, particularly given that the authors state that a very large number of participants buy no alcohol at all (i.e., outcomes are zero-inflated). From the results reported, we don't actually know to extent to which the intervention reduces overall drink purchasing, as the authors focus only on reductions in alcoholic drinks purchased among those who purchased any alcohol. That is an interesting outcome, but I do not think it is the outcome of most relevance to public health. I suggest the authors present combined marginal effects from the two parts of the model, rather than only focusing on effects among those who purchased any alcohol. 

2. I found the intervention interesting and appreciate the naturalistic randomized design used. I also appreciated that the authors are clear that the study largely offers a "proof of concept" that this type of intervention can affect consumer behavior. However, I was left wondering how important this proof of concept is given that consistent impacts were observed only for the condition that was very high in non-alcoholic drinks (75%). Although I agree that perhaps such change is in theory possible in a limited number of settings that explicitly cater to health-oriented consumers, it strikes me as wholly unrealistic for nearly all retail settings even in the medium term. The authors do not indicate any policy levers are available for mandating such a change. I think the manuscript would be much stronger if the authors could provide a stronger rationale for why it might be feasible to expect such an intervention could be scaled to a meaningful degree. 

Minor comments:

3. General: I found it very hard to remember what the condition names referred to. I suggest you rephrase the names of the conditions so that readers can immediately remember what they refer to vs. needing to refer back to the methods to remember that "High proportion" means High proportion of non-alcoholic drinks, i.e., 75% non-alcoholic/25% alcoholic drink mix. 

4. Abstract: It is not clear what "model results" refers to here, as no modeling has been introduced. As noted above, I also do not think it is appropriate to only report results from half of the hurdle model. 

5. Introduction: It would be very helpful to have a sense of current availability of non-alcohol drinks in a typical supermarket or beverage store setting. It was hard to understand what the proportions tested in the trial mean without a sense of what proportion of beverages in the current marketplace are alcoholic vs. non-alcoholic. It would likewise be helpful to have a sense of current absolute sales of these drinks (you note that sales of non-alcohol drinks are increasing, but do not state the base level). 

6. Methods: It would be helpful to have a better understanding of the differences between the simulated online store and Tesco.com. For example, how many beverages does Tesco.com offer? What proportion of those drinks are alcoholic? 

7. Methods: It struck me that soft drinks might not be a typical substitute for alcoholic beverages. Studies of soda taxes, for example, find that soda and alcohol are not strong substitutes. Could the authors state why soft drinks were used as a non-alcoholic option? If soft drinks were appropriate here, why not have included a whole range of non-alcoholic beverages that consumers might regularly purchase, like milk, or 100% juice? Offering soft drinks feels like it could have enhanced the impact of the intervention over what might be expected in the real world, as it would provide consumers with number of drink options they might not typically see near the alcoholic drinks in a store. 

8. Methods: Suggest including in the main text a few more details on how drinks were chosen for the simulated online store. The supplement makes it clear that considerable thought went into choosing these products - a few of these details in the main text would be helpful. Likewise, I suggest indicating that the products were shown in random arrangement in the main text (vs. only mentioning in the supplement). 

9. Methods: Suggest you provide more details on what instructions where given to participants for each part of the experiment. (E.g., the text states that participants were shown instructions, but does not provide details on exactly what participants were asked to do). 

10. Methods: You might consider adding a sentence that explicitly differentiates "units" from "drinks." I realize you define "units" in the primary outcome paragraph, but the secondary outcomes all sound so similar to one another that it was hard to remember the difference between "drinks" and "units," especially because "drinks" was never defined. Perhaps my confusion was partly because "drinks" can often mean "standard drinks" (at least in the US context) and units can often refer to individual products (e.g., "unit sales") regardless of how many standard drinks the product contains. Another idea would be to use the word "standard drink" instead of "unit," which (at least in the US context) might be more familiar to readers. 

11. Statistical analyses: Why were different thresholds set for the different comparisons? How were these levels selected vs. a more standard adjustment? (E.g., Bonferroni-Holm, Benjamini-Hochberg)

12. Statistical analyses: How were participants who did not buy any drinks (either alcoholic or non-alcoholic) treated in each of the analyses? 

13. Results, throughout: I found the results reporting confusing - it appears as though the model was used to calculate differences between groups in counts, but then the 95% CI's in the text are reported in in terms of proportional changes. It's not clear where these CIs come from - suggest clarifying in the analysis section or reporting CIs in the same units as the coefficients were estimated (i.e., counts). This applies throughout (including the abstract). 

14. Page 8 - Primary outcomes, "there were no differences in zero values for the other comparisons" - suggest rephrasing for clarity, e.g., to "there were no differences in likelihood of selecting no alcoholic drinks containing alcohol."

15. Page 8-9, Purchasing: It was surprising that the Higher proportion group selected a lower proportion of alcohol drinks than the lower proportion group (43% vs. 60%) and that the Lower proportion and Same proportion groups differed (60% vs. 56%) but the difference between the Higher proportion group and the Same proportion group was not significant (43% vs. 56%), despite being much larger in magnitude than the difference between Lower and Same groups. The 95% CI's reported do not overlap (which I understand is not the same as the difference between the groups being significant, though often when CIs do not overlap, there is a significant difference). Could the authors speculate on why this pattern of results might have occurred? 

16. Page 9, Per Protocol Analysis - I did not quite understand what this meant: "Of the 422 participants who purchased drinks, 344 participants purchased the drinks they had selected in the selection task (with 78 participants missing one or more drinks); 182 participants purchased exactly the drinks they had selected with no additional drinks." I think this means that the 344 purchased the drinks they selected and also additional drinks and that 78 of those 344 had missing drinks, but it was not totally clear. 

17. Page 9, Per Protocol Analysis - The authors note that although attrition was greater among participants with higher baseline alcohol purchasing, "modeling suggested this did not bias comparison between groups." This finding is crucial to the study interpretation, so I would suggest describing the results of the modeling in more detail, so readers can assess for themselves the evidence for/against biased attrition, without having to consult the supplemental materials. Additionally, in the supplemental table S4B, it's clear that the interaction between drinks and remained/lost is significant for nearly all results, especially when using Model A. This should be addressed in both the results and the discussion of the main text. 

18. Table 3. It was unclear what the coefficients reported for part 1 of the hurdle model referred to. I expected to see differences in predicted probability of selecting any alcohol, or perhaps an odds ratio, but it appears as if counts are reported. I would also suggest adding a table note describing the modeling in a bit more detail and clarifying where each estimate comes from (e.g., for the primary outcome, it was not clear where the 95% CIs for the proportional reductions come from for part 2 of the model given that a count model was used, and likewise it took me a minute to understand what the p-values refer to). 

19. Figure 1: For Group 1, 1 participant was excluded from analysis for not meeting drinking inclusion criteria…Shouldn't this person have been excluded prior to enrollment/randomization? 

20. Figure 1: In the allocation boxes, I would suggest keeping the n's for "did not receive full intervention" in same bullet point as their explanation (vs. 2 separate bullet points)

21. Figure 2: Do you have any data on the participants who dropped out in the selection task? I realize the dropout rate is not very high, and does not appear to be highly differential by arm (though was nearly 2x as high in the 50/50 alcoholic-non-alcoholic arm), but if you have any screening data on these participants, it might be useful to see whether the people who dropped out in each arm were roughly similar to one another 

22. Supplemental S2: Some n's are missing in the column headings. 

23. Supplemental S4B: I did not understand what was being reported in this table - suggest adding additional table notes, particularly given that these analyses are not described in detail in the main text. Additionally, it's not clear what "poor diagnostics" means in this context. This feels especially important given that the interaction was significant for most Model A results. 

*** Reviewer #2: 

Thank you for the opportunity to review this paper, which examines the impact of the proportion of alcohol drinks on alcohol units selected. The research question is valuable and the topic is timely given the high public health burden of alcohol consumption. However, my enthusiasm for the research is limited mostly by concerns about the external validity of the simulated "store." 

Major concerns:

* The authors describe the simulated "store" as a naturalistic setting but it appears to be images of products mocked up in Qualtrics. In my opinion, this protocol does not feel very realistic and no data are provided about the external validity of the platform. I have questions such as: have the authors used this protocol before? Does behavior in the Qualtrics store correlate with actual shopping behavior? What did the "store" look like to participants? (I didn't see any images in the paper). Were prices displayed? Were there any elements of a real online store like price promotions, banners, etc.? What kinds of information about the products were available within the simulated store? (e.g., could participants view nutrition information about the products?) 

* I appreciate that the authors attempted to heighten the realism of the protocol by having participants "purchase" their selected drinks in a real store. However, I am not convinced that this element of the protocol really addresses the concerns with external validity because the real store did not reflect their experimental arm (proportion of alcoholic drinks) and many more products were available than in the initial task. Ultimately, the protocol felt a bit clunky and hard to follow and the case for doing the second task felt fairly weak to me. 

* For the power analysis - I wondered why the authors did not attempt to estimate a range of plausible effect sizes based on research on product availability from other domains (e.g., food). As written, the study feels exploratory in nature. 

* The authors briefly comment on the fact that some of the non-alcoholic options contained sugar. I would encourage the authors to more thoroughly address the potential unintended consequences of people switching from alcohol to sugar-sweetened beverages given different but substantial health risks to drinking SSBs. 

Other comments

* I wondered how the authors arrived at the ranges of the proportions they manipulated (75%, 50%, 25%)? 

* More information on how the authors picked the drink options would be helpful. Was this based on any data reflecting popular choices / market share / sales? 

* More detail on the scalability of this intervention would be useful in the intro and discussion. Is this intervention something that retailers might do voluntarily? If so, is there precedent for stores doing this? Or is this intervention something that would be required as a policy at the local or federal level? (and similarly if so, is this on the table as part of policy discussions/plans anywhere?)

* Were any of the analytic decisions not pre-registered? Outlining deviations / modifications to the protocol (if any) would be helpful. For example, the switch to the hurdle model seems like it was not pre-specified based on my review of the OSF plan. 

* I would suggest including raw percentages in the text of the results in the primary outcome section for the part I of the hurdle model. 

* Does the p-value adjustment use a specific approach? What is the rationale for dividing 5%/2? 

*** Reviewer #3: 

The article is more cautious now, and I recommend publication

*** Reviewer #4: 

Overall, I found the manuscript clear and easy to follow. I have only have minor comments listed below.

I note that no adjustment was made for multiplicity (3 arms) in the sample size calculation; however, according to the SAP (hypothesis testing section), the two main comparisons are set using alpha of 2.5%

Please confirm that the criteria used to exclude data (e.g. incomplete/suspicious data or forged receipt) was defined and applied while unaware of the group allocation.

The SAP pre-specifies that a generalised linear model would be used. I understand the issue of skewness and excess of zero which led to the use of a two-part model; however, I would be keen to see the results using a linear model including all data (i.e. including zeros) as initially planned. This could be reported in the supplement. 

I find Table 3 difficult to read. I would recommend adopting the following format: mean difference (95% CI), p=0.xxx. For example: -0.64 (-1.44; 0.17) p=0.121. The same applies to Table 4 and tables in the supplement.

When reporting outcome results in the text, please report the main results as mean differences instead of percentage changes. In Tables, please also consider removing the reporting of results as percentage change. Percentage changes could potentially be reported in the supplement only and used in the discussion to assist with the interpretation. 

When reporting the results for the primary outcome please consider first reporting the proportion of participants selecting at least one alcoholic drink (part 1 of the model) before reporting the results of part 2.

422/640 (66%) of participants went on to purchase drinks. Are the reasons for not proceeding to the purchasing stage known? If so, please consider reporting them. It would also be interesting to further understand whether the profiles of participants differ according to whether or not they went on to purchase drinks. Pease compare the characteristics (both baseline measures and purchasing intention) of participants who carried on with the purchase vs those who did not. In case of meaningful differences, please consider additional sensitivity analyses.

I understand that the analysis models were unadjusted. I would suggest performing sensitivity analyses adjusted for key prognostic factors such as alcohol consumed the previous week (or alcohol purchased and the previous week or AUDIT score).

-Laurent Billot

***

[LINK]

---

## [Decision Letter · Decision Letter 2]

19 Jan 2023

Dear Dr. Clarke,

Thank you very much for re-submitting your manuscript "Impact on alcohol selection and online purchasing of changing the proportion of available non-alcoholic versus alcoholic drinks: A randomised controlled trial" (PMEDICINE-D-22-01553R2) for review by PLOS Medicine.

I have discussed the paper with my colleagues and it was also seen again by 3 reviewers. I am pleased to say that provided the remaining editorial and production issues are dealt with we are planning to accept the paper for publication in the journal.

[LINK]

We look forward to receiving the revised manuscript by Jan 26 2023 11:59PM.   

Sincerely,

Philippa Dodd MBBS MRCP PhD 

PLOS Medicine

plosmedicine.org

Requests from Editors:

GENERAL

Thank you for your very considerate and detailed responses to previous reviewer and editor comments.

Please see below for further minor editorial revisions

ABSTRACT

Line 16: “…16 purchasing. 607 participants (60% female, mean age = 38 years)…” please include the age range

Line 17: “. In the first part of a hurdle model, a greater proportion…” suggest reporting the results followed by the statistical method used to derive them, “ A greater proportion….as demonstrated by the first part of…” similarly for line 21

Thank you for including p-values and confidence intervals. Please define CI at first use.

Throughout the abstract (and main text) p-values are presented for some outcome measures and confidence intervals for others (in the main text CIs are absent – see below). Please ensure that for all outcome measures both p-values and 95% CIs are reported. 

suggest placing CIs within square parentheses as follows: “(95% CI [15.24, 19.68], p= or p<0.001)

When reporting p-values please report as <0.001 (not <.001) and where higher please report actual values as p= 0.12, for example, not p= .12 Please check and amend throughout including the main manuscript text, tables and figures, where relevant

AUTHOR SUMMARY

Thank you for including an author summary.

It’s rather difficult to appreciate exactly what you did just by reading the author summary alone and this should not be the case.

Line 6 onwards – it might be helpful to include some of the details you report early in your introduction to give context to the importance/necessity of the study

Line 15 onwards - detail is missing which explains clearly to the reader what you did, how you did it and how doing it is helpful. Please revise accordingly - some specific points are detailed below.

Please include the study design and number of participants “In a randomised controlled trial of X number of ppts…” for example, or something similar

It may be beneficial to the reader to further elaborate here a) on the simulation part of the study, what was simulated, what did participants look at? suggest explaining briefly what “availability” interventions are to give context and clarity and b) separately describe what participants did in the actual supermarket and how the two events are connected, as written its not clear, a lot is left to assumption (or knowledge of detail contained within the main manuscript)

Line 16: at “Participants…” please make into a separate bulleted point

Line 21: Additional detail of your findings would be helpful here also – reducing the amount of alcohol how and when

Line 26: suggest including an additional point which states the implications of this i.e. why is that helpful?

INTRODUCTION

Line 47: Suggest removing/substituting the word “Although…” or at line 46

METHODS and RESULTS

When reporting p-values please report as <0.001 (not <.001) and where higher please report actual values. Please check and amend throughout including the main manuscript text, tables and figures, where relevant

For all outcome measures please report p-values and 95% CIs, as detailed above

Please define %ABV at first use (line 49 was the 1st I could find but please check)

TABLES

Please ensure that each table is affiliated to an appropriate caption/legend which clearly details the table contents, without the need to refer to the main manuscript text. Please check and amend throughout including in the supplementary files

Please ensure that p-values as well as 95% CIs are reported in all tables. In the table captions/legends where p-values are reported please define the test used to derive them and the significance level

In the table titles please ensure that the abbreviation CI is defined

Please report p as <0.001 (not .001) or if higher the exact p-value as p = 0.14 (not .14), for example

Please check and amend throughout

DISCUSSION

Please remove all sub-headings from the discussion, including the conclusion such that it reads as a single piece of continuous prose

Page 12, line 26: Suggest tempering this statement “Our data show that

Page 12, 37: “The findings of the current study are in line with…” suggest “consistent with…” 

Page 12, line 45: Please add “to the best of our knowledge...” or something similar

Page 14, line 46: Suggest “To our knowledge, …” or “This randomised controlled trial is the first, that we are aware of, to assess…” or something similar

REFERENCES

Please check the bibliography to ensure that journal name abbreviations are those found in the National Center for Biotechnology Information (NCBI) databases. 

SOCIAL MEDIA

To help us extend the reach of your research, please provide any Twitter handle(s) that would be appropriate to tag, including your own, your coauthors’, your institution, funder, or lab. Please enter any handles you wish to be included into the relevant part of the manuscript submission form.

Comments from Reviewers:

Reviewer #1: Thank you for the opportunity to review the revised manuscript, Impact on alcohol selection and online purchasing of changing the proportion of available non-alcoholic versus alcoholic drinks: A randomised controlled trial. The authors have responded fully and thoughtfully to each of my comments. I have no further questions. 

Reviewer #2: I thought the reviewers were very responsive to comments. No further comments from me! 

Reviewer #4: My comments have been adequately addressed. It is good to see both parts of the models reported together with the marginal effects. I believe it makes the interpretation much clearer.

-Laurent Billot

[LINK]

---

## [Editor Report · Decision Letter 3]

8 Feb 2023

Dear Dr Clarke, 

On behalf of my colleagues and the Academic Editor, Dr. Anna H Grummon, I am pleased to inform you that we have agreed to publish your manuscript "Impact on alcohol selection and online purchasing of changing the proportion of available non-alcoholic versus alcoholic drinks: A randomised controlled trial" (PMEDICINE-D-22-01553R3) in PLOS Medicine.

Before your manuscript can be published we require the following:

1) Please ensure you include the URL instead of the placeholders “(DOI [XXX] and OSF link [XXX])” when you re-submit your manuscript. This is a journal requirement for publication.

2) Please include the study protocol document and analysis plan, with any amendments, as Supporting Information to be published with the manuscript. This is a journal requirement for publication. I could not locate these in my version of the manuscript but my apologies if I have missed it. 

3) Please be reminded to upload your preferred twitter handles to the relevant part of the manuscript submission form, if you have not already done so.

PRESS

Sincerely, 

Philippa Dodd, MBBS MRCP PhD  

PLOS Medicine